# The human visual system differentially represents subjectively and objectively invisible stimuli

**Timo Stein**[1]*, **Daniel Kaiser**[2], **Johannes J. Fahrenfort**[1,3], **Simon van Gaal**[1]

1 Brain and Cognition, Department of Psychology, University of Amsterdam, the Netherlands, 2 Department of Psychology, University of York, United Kingdom, 3 Cognitive Psychology, Department of Experimental and Applied Psychology, Vrije Universiteit Amsterdam, the Netherlands

* timo@timostein.de

**Data Availability Statement:** All quantitative observations that underlie the data summarized in the figures and results of our paper are available on OSF (https://osf.io/qus5v/).

## Abstract

The study of unconscious processing requires a measure of conscious awareness. Awareness measures can be either subjective (based on participant's report) or objective (based on perceptual performance). The preferred awareness measure depends on the theoretical position about consciousness and may influence conclusions about the extent of unconscious processing and about the neural correlates of consciousness. We obtained functional magnetic resonance imaging (fMRI) measurements from 43 subjects while they viewed masked faces and houses that were either subjectively or objectively invisible. Even for objectively invisible (perceptually indiscriminable) stimuli, we found significant category information in both early, lower-level visual areas and in higher-level visual cortex, although representations in anterior, category-selective ventrotemporal areas were less robust. For subjectively invisible stimuli, similar to visible stimuli, there was a clear posterior-to-anterior gradient in visual cortex, with stronger category information in ventrotemporal cortex than in early visual cortex. For objectively invisible stimuli, however, category information remained virtually unchanged from early visual cortex to object- and category-selective visual areas. These results demonstrate that although both objectively and subjectively invisible stimuli are represented in visual cortex, the extent of unconscious information processing is influenced by the measurement approach. Furthermore, our data show that subjective and objective approaches are associated with different neural correlates of consciousness and thus have implications for neural theories of consciousness.

## Introduction

Determining the function and neural correlates of human consciousness is one of the most challenging topics in psychology and cognitive neuroscience today [1–3]. The scientific study of consciousness requires pitting conscious processes against comparable unconscious processes [4]. One powerful approach is to compare neural processing between stimuli presented outside conscious awareness and stimuli that are consciously perceived. Although all major theories of consciousness are based on the notion that stimuli can be processed unconsciously

**Funding:** This project has received funding from the European Research Council (ERC) under the European Union's Horizon 2020 research and innovation program (European Research Council starting grant 715605 awarded to SvG), and from the German Research Foundation (DFG, KA4683/2-1 to DK). The funders had no role in study design, data collection and analysis, decision to publish, or preparation of the manuscript.

**Competing interests:** The authors have declared that no competing interests exist.

**Abbreviations:** BF, Bayes factor; EPI, echo-planar imaging; FDR, false discovery rate; FFA, fusiform face area; fMRI, functional magnetic resonance imaging; GLM, general linear model; LOC, lateral occipital complex; MVPA, multivoxel pattern analysis; obj-inv, objectively invisible; obj-vis, objectively visible; OFA, occipital face area; OPA, occipital place area; PAS, perceptual awareness scale; PPA, parahippocampal place area; ROI, region of interest; SDT, signal detection theory; subj-inv, subjectively invisible; subj-vis, subjectively visible; V1, primary visual cortex.

[5–8], the scope and extent of unconscious processing are highly debated, with estimates ranging from low-level perceptual analysis [9,10] to high-level object categorization [11–13] and full-blown unconscious cognition and reasoning [14,15]. One important cause of this controversy relates to theoretical disagreements about how to measure consciousness and how to demonstrate absence of conscious awareness [11,16].

The most intuitive approach is to simply ask participants to introspectively report their experience of a barely perceivable (e.g., masked) stimulus [17]. Recent studies adopting such subjective awareness measures found that visual stimuli reported as "invisible" still undergo high-level processing. For example, subjectively invisible stimuli are encoded [18,19] and stored [20–22] in visual cortex, enabling above-chance perceptual discrimination even after a memory delay [23,24]. By linking behavioral performance for subjectively invisible stimuli to brain activity, the neural correlates of such "blindsight"-like performance can be recorded in normal observers [25]. Subjective measures assume that an observer's decision to report awareness accurately distinguishes the presence versus absence of conscious perceptual information. However, this assumption is at odds with research on perceptual decision-making, which shows that subjective measures do not accurately reflect perception, because they are susceptible to decision biases [26,27]. Participants often have a conservative bias [28], such that an "invisible" response merely indicates that a stimulus was relatively difficult to see [29]. Thus, subjective measures may misclassify (partially) conscious stimuli as unconscious. Studies adopting purely subjective measures therefore most likely overestimate the extent of unconscious processing [26,30,31].

To convincingly rule out conscious perception, objective awareness measures based on performance are required [26]. Crucially, these measures should target the key stimulus characteristic of interest [32]. For example, when contrasting neural responses evoked by masked faces and houses, participant's performance in discriminating the two stimulus categories should not exceed what is expected by chance [33]. However, this approach is also riddled with challenges. First, statistically, scientists have taken a failure to reject the null hypothesis (of chance performance, $p > 0.05$) as support for the null hypothesis, which is invalid in the standard frequentist hypothesis testing framework. A nonsignificant effect can be related to measurement noise and lack of statistical power rather than genuine invisibility (chance performance). In neuroimaging studies in particular, this represents a serious concern as sample sizes tend to be very small (e.g., only four [34], five [33,35], six [36], seven [37], or eight [38] participants). Further, objective awareness is often only measured "offline" in a block separate from the functional magnetic resonance imaging (fMRI) recordings (or even outside the MRI scanner) [35–42], and with considerably fewer trials [35,37,39–41] and sometimes also fewer participants [38,40]. Given these issues, it is unknown how stimuli that are genuinely invisible according to the objective definition are represented in different cortical areas.

Here, our first goal was thus to measure the extent of unconscious processing of objectively invisible stimuli in human visual cortex. To ensure sufficient statistical power, we included data from 43 human observers who completed a large number of trials while measuring brain activity evoked by masked faces or houses using a multiband fMRI sequence with fast acquisition time and high spatial resolution. An "online" (trial-by-trial) measure of perceptual discriminability of faces and houses served as an exhaustive [32,43] measure of objective awareness, leading to equal numbers of trials to calculate objective discrimination performance and to evaluate the extent of neural processing of masked face/house stimuli. Perceptual sensitivity was analyzed with Bayesian statistics to establish genuine objective invisibility.

Compared to subjective invisibility, objective invisibility requires strong reduction of stimulus strength, e.g., by brief presentation times, low contrasts, or strong masking, which reduces neural responses. Estimates of unconscious processing, both in behavior and in neural

recordings, may thus critically depend on selection and implementation of method and statistics for establishing absence of conscious awareness. However, as objective and subjective awareness measures have not been compared in the same study, it is unclear whether the 2 approaches are merely associated with quantitative differences (e.g., subjective measures allowing for stronger neural responses to unconscious stimuli) or whether important qualitative differences exist (e.g., subjective measures allowing for unconscious processing of distinct higher-level stimulus properties).

Hence, our second goal was to directly compare the two approaches. We used fMRI to measure brain activity evoked by masked faces or houses that were visible, subjectively invisible, or objectively invisible. In the objective condition, visibility was controlled experimentally by fixing the contrast of the masks. In the subjective condition, visibility was controlled by the participant's response (i.e., visibility became the dependent variable), and mask contrast was continuously adjusted to yield similar proportions of trials rated as subjectively visible and invisible. This approach yielded a similar number of trials for objectively and subjectively invisible stimuli, resulting in similar statistical power to detect neural effects in objective and subjective conditions. Using multivariate pattern analysis, we tracked neural representations of faces and houses along the visual processing hierarchy, from early visual cortex to object-selective lateral occipital complex (LOC [44,45]), as well as in category-selective regions in the lateral occipital (occipital face area [OFA] [46] and occipital place area [OPA] [47]) and ventrotemporal cortices (fusiform face area [FFA] [48] and parahippocampal place area [PPA] [48]). This analysis allowed us to establish the level of representation of objectively and subjectively invisible stimuli in human visual cortex.

## Results

On every trial, participants discriminated between faces and houses (objective measure, face/house) and simultaneously indicated stimulus visibility (subjective measure, visible/invisible). Participants were asked to be as accurate as possible, guessing when necessary. For the subjective measure, instructions emphasized that participants should press "visible" even when they had only a vague idea of the stimulus category and press "invisible" only when they had absolutely no idea of the stimulus category. Faces and houses were presented for 16.7 ms, sandwiched between masks (**Fig 1A**). Subjective stimulus visibility was based on the participant's response, while objective stimulus visibility was controlled by the experimenter. In the subjective condition (50% of trials), mask contrast was adjusted through an adaptive 1-up 1-down "staircase" procedure. Following a "visible" response, mask contrast was increased by 4%. Following an "invisible" response, contrast was lowered by 4%. This continuous adjustment was intended to yield a similar number of subjectively visible (subj-vis) and subjectively invisible (subj-inv) trials. This allowed us to test the effect of subjective visibility with minimal differences in mask contrast (subj-vis: mean contrast 7.8%, SD 4.4; subj-inv: mean contrast 11.9%, SD 4.3). In the objective condition (50% of trials), mask contrast was either low (2%) to achieve clear stimulus visibility (objectively visible (obj-vis) condition) or high (100%) to achieve chance-level discrimination (objectively invisible (obj-inv) condition).

### Masking efficiency experiment

We first ran a separate behavioral experiment to measure the influence of mask contrast on perceptual discriminability and to determine optimal mask contrast for objective invisibility ($N = 17$, **Fig 1B**). Face/house discriminability and subjective visibility (**Fig 1C**) increased similarly with decreasing masking strength ($F_{(8, 128)} = 247.15$, $p < 0.001$, $\eta_p^2 = 0.94$, $BF_{10} = 2.07 \times 10^{74}$ and $F_{(8, 128)} = 215.69$, $p < 0.001$, $\eta_p^2 = 0.93$, $BF_{10} = 8.99 \times 10^{69}$, respectively; also see

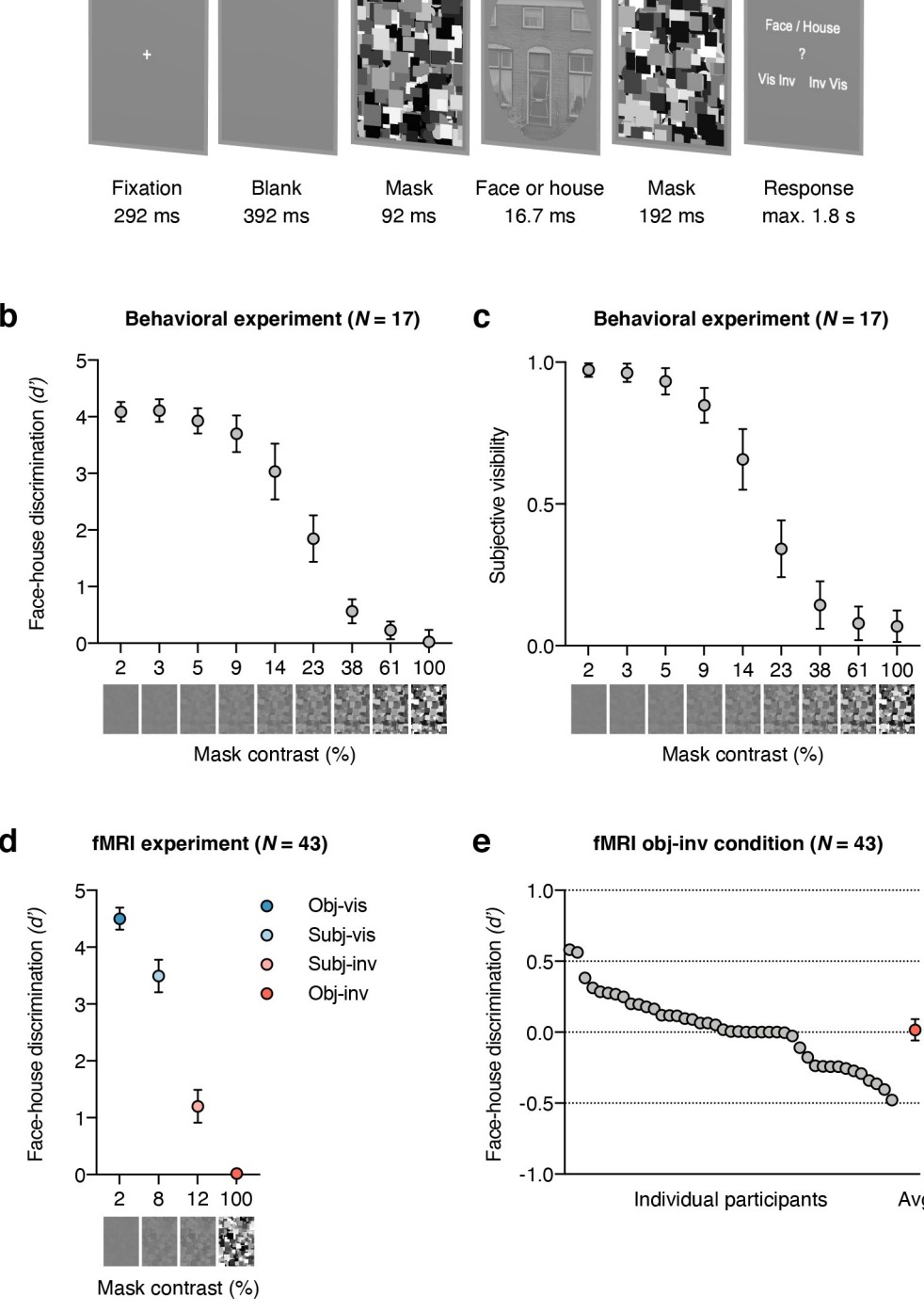

**Fig 1. Experimental paradigm, results from the behavioral masking efficiency experiment and behavioral results from the fMRI experiment. (a)** Example trial: A face or house stimulus was forward and backward masked by a mask with variable contrast, depending on the visibility condition. On every trial, participants judged both the category and visibility of the masked stimulus, yielding measures of objective category discriminability and of subjective stimulus visibility. **(b)** Results from the behavioral masking efficiency experiment. Mean face-house discriminability is plotted for 9 different masking contrast levels (on a $\log_{10}$ scale). Only with 100% masking contrast did discriminability not significantly exceed chance performance. **(c)** Mean visibility ratings from the behavioral masking efficiency experiment. **(d)** Behavioral results from the fMRI experiment. Mean face-house discriminability is shown for the four

visibility conditions. Mask contrast was set to 2% in obj-vis and to 100% in obj-inv; in the subjective conditions, mask contrast was continuously adjusted by an adaptive staircase procedure, which resulted in a mean contrast (across the group) of 8% in subj-vis, and of 12% in subj-inv. Discrimination performance was significantly above chance for subjectively invisible stimuli (subj-inv) but not for objectively invisible stimuli (obj-inv). All error bars represent 95% confidence intervals; in (d), for the obj-inv condition, the error bar was smaller than the symbol. **(e)** Individual participant's face-house discrimination performance in the obj-inv condition (in *d'*). Every gray circle represents a participant, and the red circle shows the group mean with its 95% confidence interval. Data underlying this figure are available on OSF (https://osf.io/qus5v/). fMRI, functional magnetic resonance imaging; obj-inv, objectively invisible; obj-vis, objectively visible; subj-inv, subjectively invisible; subj-vis, subjectively visible.

**S1 Fig**). A set of one-tailed *t* tests showed significant above-chance performance for all mask contrasts (all $p < 0.005$, all $d_z > 0.75$, all $BF_{+0} > 14$), except for 100% mask contrast (M = 0.03, SD = 0.41, $t_{(16)} = 0.26$, $p = 0.40$, $d_z = 0.06$, $BF_{0+} = 3.26$). Bayes factor ($BF_{0+}$) (note that BF was calculated in JASP [93] with default prior scales [Cauchy distribution, scale 0.707]; see S1 Text and S2 Fig for additional analyses of discrimination performance in the obj-inv condition) indicated that the null hypothesis of chance performance was about three times more likely than the alternative hypothesis of above-chance performance, which represents "moderate" evidence for the null hypothesis [49]. Thus, full mask contrast was necessary to achieve objective invisibility and was therefore used in the following fMRI experiment. Note that optimal mask contrast was determined based on the group data from the masking efficiency experiment and set accordingly for the whole group of fMRI participants. An alternative approach that is sometimes thought to be better suited to localize the "sweet spot" for unconscious processing is to calibrate mask contrast individually to allow for maximum signal strength for every participant [50]. However, the masking efficiency experiment revealed low between-subject variability, as well as low reliability for the estimation of individual performance measures for high mask contrasts (see **S1 Text** and **S1 Table**). We thus considered group-based calibration of mask contrast better suited to ensure chance performance for all participants in the fMRI experiment, although we cannot exclude the possibility that our group-based calibration approach did not result in optimal stimulus strength for every individual observer. This might have reduced the chances of finding higher-level processing for objective invisibility and might have inflated the difference between subjective and objective invisibility in our study.

## Behavior

In the fMRI experiment (*N* = 43), face/house discriminability *(d')* was high in obj-vis (M = 4.50, SD = 0.63) and in subj-vis trials (M = 3.50, SD = 0.93) and remained above chance in subj-inv (M = 1.20, SD = 0.94, *t* test against chance, $t_{(42)} = 8.36$, $p < 0.001$ (one-tailed), $d_z = 1.26$, $BF_{+0} = 8.59 \times 10^9$, **Fig 1D**. Such above-chance performance for subjectively invisible stimuli is often referred to as a blindsight-like phenomenon of unconscious stimulus processing. Importantly, in obj-inv trials, discrimination performance (M = 0.02, SD = 0.25) did not differ significantly from chance (**Fig 1E**), with moderate evidence for the null hypothesis of chance-level discrimination ($t_{(42)} = 0.45$, $p = 0.33$ (one-tailed), $d_z = 0.07$, $BF_{0+} = 4.13$; also see **S2 Fig**). Regarding subjective visibility, participants reported that the stimulus was visible in 94.6% (SD 5.5) of the obj-vis trials and invisible in 96.3% (SD 5.1) of the obj-inv trials. In the subj-vis and subj-inv trials, mean subjective visibility was by definition 100% and 0%, respectively, because these trial categories were conditioned on the subjective visibility response.

## Category-specific information in visual cortex

We tested in which subregions of visual cortex activation patterns reliably distinguished between faces and houses. To obtain a measure of category information in the neural

responses, activity patterns were correlated with "benchmark" patterns for unmasked faces and houses obtained in an independent localizer scan where participants did a simple one-back task on stimuli presented in a block design. This approach is statistically powerful and rules out the possibility that multivoxel pattern correlations reflect cognitive processes related specifically to the task or decision and motor processes (as these differed between the localizer and the main experiment) [51,52]. Category information was quantified as the difference between within- and between-category multivoxel pattern correlations ($\Delta r$) between the independent localizer and every visibility condition in the main experiment for 4 separate regions of interest (ROIs, see **Fig 2**). Note that differences in multivoxel pattern correlations can result from changes in the fine-grained activation patterns within the face- and house-selective regions or from univariate activation differences between these regions (for complimentary univariate analyses, see **S1 Text** and **S3 Fig**).

First, we examined activity patterns in primary visual cortex (V1) and object-selective visual cortex (LOC [44]). There were marked differences between the regions, with overall greater category information in LOC than in V1 ($F_{(1, 42)} = 260.39$, $p < 0.001$, $\eta_p^2 = 0.86$, $BF_{10} = 4.53 \times 10^{31}$). Category information also differed between visibility conditions ($F_{(3, 126)} = 79.03$, $p < 0.001$, $\eta_p^2 = 0.65$, $BF_{10} = 1.58 \times 10^{28}$), and these differences were more pronounced in LOC than in V1 (interaction, $F_{(3, 126)} = 99.86$, $p < 0.001$, $\eta_p^2 = 0.70$, $BF_{10} = 1.48 \times 10^{30}$). To test whether activity patterns discriminated between faces and houses with above-chance accuracy, separate one-tailed $t$ tests were carried out for every region and every visibility condition. V1 carried significant category information in obj-vis ($t_{(42)} = 3.44$, $p < 0.001$, $d_z = 0.52$, $BF_{+0} = 46.73$), subj-vis ($t_{(42)} = 3.97$, $p < 0.001$, $d_z = 0.60$, $BF_{+0} = 188.02$), and obj-inv ($t_{(42)} = 2.55$, $p = 0.007$, $d_z = 0.39$, $BF_{+0} = 5.74$), but not in subj-inv ($t_{(42)} = 1.12$, $p = 0.135$, $d_z = 0.17$, $BF_{0+} = 1.97$). Activity patterns in LOC discriminated between faces and houses with above-chance accuracy in all visibility conditions (obj-vis ($t_{(42)} = 16.69$, $p < 0.001$, $d_z = 2.55$, $BF_{+0} = 2.43 \times 10^{17}$), subj-vis ($t_{(42)} = 12.88$, $p < 0.001$, $d_z = 1.96$, $BF_{+0} = 3.52 \times 10^{13}$), subj-inv ($t_{(42)} = 5.21$, $p < 0.001$, $d_z = 0.80$, $BF_{+0} = 7.22 \times 10^3$), and also in obj-inv ($t_{(42)} = 3.28$, $p = 0.001$, $d_z = 0.50$, $BF_{+0} = 31.19$). Thus, LOC contained category information for both subjectively and objectively invisible stimuli.

To gain insight into the level of representation in higher-level visual cortex, we examined activity patterns in posterior versus anterior category-selective regions, which are partially overlapping with the spatially extended LOC region (**Fig 2**). The 2 more posterior areas represent more basic and "local" visual aspects of faces and houses (OFA [46] and OPA [47,53]), whereas the two more anterior areas also contain "global" and abstract categorical representations (FFA and PPA [48]). Overall, anterior regions carried more category information than posterior regions ($F_{(1, 42)} = 47.09$, $p < 0.001$, $\eta_p^2 = 0.53$, $BF_{10} = 4.53 \times 10^{31}$). There were also significant differences between visibility conditions ($F_{(3, 126)} = 63.76$, $p < 0.001$, $\eta_p^2 = 0.60$, $BF_{10} = 1.58 \times 10^{28}$) and a significant interaction ($F_{(3, 126)} = 26.94$, $p < 0.001$, $\eta_p^2 = 0.39$, $BF_{10} = 1.48 \times 10^{30}$). Activity patterns in the posterior areas (OFA/OPA) discriminated between faces and houses in all conditions (obj-vis [$t_{(42)} = 10.55$, $p < 0.001$, $d_z = 1.61$, $BF_{+0} = 6.99 \times 10^{10}$], subj-vis [$t_{(42)} = 6.99$, $p < 0.001$, $d_z = 1.07$, $BF_{+0} = 1.72 \times 10^6$], subj-inv [$t_{(42)} = 3.90$, $p < 0.001$, $d_z = 0.59$, $BF_{+0} = 155.49$], and obj-inv [$t_{(42)} = 2.72$, $p = 0.005$, $d_z = 0.42$, $BF_{+0} = 8.27$]). Also in the anterior areas (FFA/PPA) category information was significant in all visibility conditions. Strong evidence for above-chance discrimination of faces and houses was obtained only for obj-vis ($t_{(42)} = 11.43$, $p < 0.001$, $d_z = 1.74$, $BF_{+0} = 7.89 \times 10^{11}$), subj-vis ($t_{(42)} = 9.61$, $p < 0.001$, $d_z = 1.47$, $BF_{+0} = 4.86 \times 10^9$), and subj-inv ($t_{(42)} = 2.72$, $p = 0.005$, $d_z = 0.90$, $BF_{+0} = 5.88 \times 10^4$). By contrast, for obj-inv, there was no solid evidence for above-chance discrimination in FFA/PPA, with only weak and inconsistent statistical evidence ($t_{(42)} = 1.97$, $p = 0.028$, $d_z = 0.30$, $BF_{+0} = 1.86$).

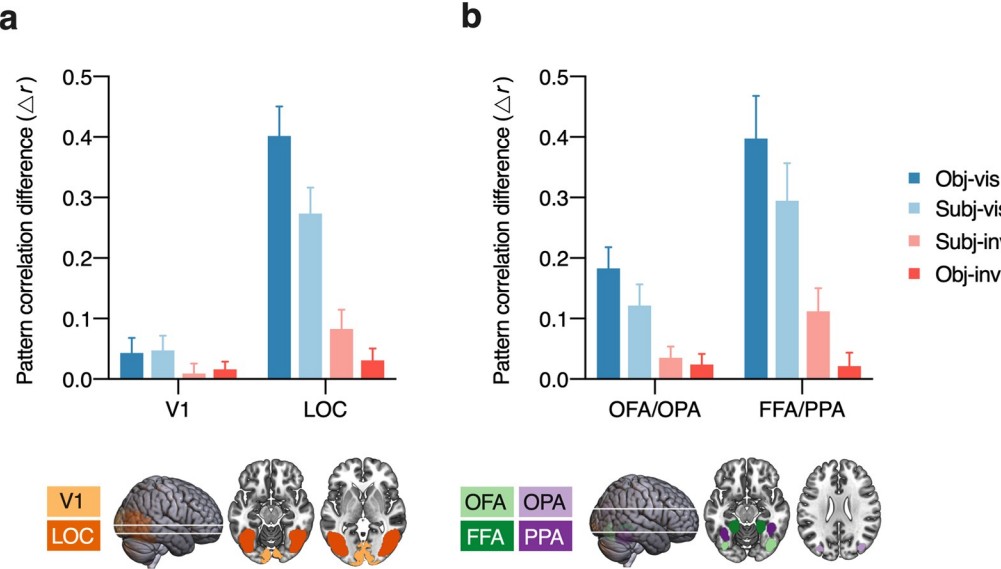

**Fig 2. Category information across regions in visual cortex.** Multivoxel pattern correlations between the 4 visibility conditions in the main experiment and the independent localizer. Bars show the mean difference between within-category and between-category multivoxel pattern correlations ($\Delta r$). Error bars represent 95% confidence intervals. **(a)** Results for early visual cortex (V1) vs. object-selective visual cortex (LOC). **(b)** Results for posterior category-selective areas (OFA/OPA) vs. anterior category-selective areas (FFA/PPA). Data underlying this figure are available on OSF (https://osf.io/qus5v/). FFA, fusiform face area; LOC, lateral occipital complex; OFA, occipital face area; OPA, occipital place area; PPA, parahippocampal place area; V1, primary visual cortex.

## Differences between visibility conditions

To directly test the effect of different definitions of (in)visibility, we compared activity patterns using paired $t$ tests (two-tailed). Both in V1 and LOC, subjective visibility was associated with greater category information than subjective invisibility (V1: $t_{(42)} = 2.85$, $p = 0.007$, $d_z = 0.43$, $BF_{10} = 5.57$; LOC: $t_{(42)} = 9.68$, $p < 0.001$, $d_z = 1.48$, $BF_{10} = 3.00 \times 10^9$), but this effect was larger in LOC (interaction, $F_{(1, 42)} = 62.93$, $p < 0.001$, $\eta_p^2 = 0.60$, $BF_{10} = 7.99 \times 10^5$), indicating a greater effect of subjective awareness in object-selective visual cortex than in early visual cortex. Also the way of establishing invisibility influenced the two regions differently (interaction, $F_{(1, 42)} = 9.02$, $p = 0.004$, $\eta_p^2 = 0.18$, $BF_{10} = 11.14$). While there was no significant difference in category information between subjectively and objectively invisible stimuli in V1 ($t_{(42)} = -0.73$, $p = 0.47$, $d_z = 0.11$, $BF_{01} = 4.72$), in LOC category information was significantly greater for subjectively than for objectively invisible stimuli ($t_{(42)} = 2.67$, $p = 0.011$, $d_z = 0.41$, $BF_{10} = 3.72$).

Similar differences were obtained for posterior versus anterior category-selective regions. Although both posterior and anterior regions carried more category information for subjectively visible than for subjectively invisible stimuli (OFA/OPA: $t_{(42)} = 4.24$, $p < 0.001$, $d_z = 0.64$, $BF_{10} = 195.43$; FFA/PPA: $t_{(42)} = 6.42$, $p < 0.001$, $d_z = 0.98$, $BF_{10} = 1.49 \times 10^5$), this effect was larger in FFA/PPA (interaction, $F_{(1, 42)} = 11.41$, $p = 0.002$, $\eta_p^2 = 0.21$, $BF_{10} = 5.66$), indicating a greater effect of subjective awareness in anterior than in posterior category-selective regions. Critically, also the method of establishing invisibility had a different effect on the two ROIs (interaction, $F_{(1, 42)} = 20.43$, $p < 0.001$, $\eta_p^2 = 0.33$, $BF_{10} = 28.18$). In OFA/OPA, there was no significant difference between subjectively and objectively invisible stimuli ($t_{(42)} = 0.79$, $p = 0.44$, $d_z = 0.12$, $BF_{01} = 4.50$), while FFA/PPA carried more category information for subjectively than for objectively invisible stimuli ($t_{(42)} = 4.23$, $p < 0.001$, $d_z = 0.64$, $BF_{10} = 195.71$).

## Posterior–anterior category information gradient

Another way to follow up on the significant visibility × ROI interactions is to test differences in category information between ROIs, separately for every visibility condition. For obj-vis, subj-vis, and subj-inv, this analysis revealed a posterior–anterior gradient, with increasing category information from V1 to LOC (all $t_{(42)} > 4.50$, $p < 0.001$, $d_z > 0.70$, $BF_{10} > 596.40$, **Fig 2A**) and from OFA/OPA to FFA/PPA (all $t_{(42)} > 4.45$, $p < 0.001$, $d_z > 0.67$, $BF_{10} > 373.24$, **Fig 2B**). For obj-inv, this gradient was not robust. Category information did not differ between V1 and LOC ($t_{(42)} = 1.70$, $p = 0.097$, $d_z = 0.26$, $BF_{01} = 1.63$) or between OFA/OPA and FFA/PPA ($t_{(42)} = 0.20$, $p = 0.842$, $d_z = 0.03$, $BF_{01} = 5.95$), indicating that processing of objectively invisible stimuli is limited to visual shape features.

## Whole-brain searchlights

The ROI analyses showed that both subjective visibility (subj-vis vs. subj-inv) and the method for establishing invisibility (subj-inv vs. obj-inv) have the strongest effects in higher-level visual cortex. To further substantiate these findings and to test their spatial specificity, we conducted "searchlight" analyses [54]. These analyses revealed areas carrying significant category information across the whole brain (corrected for multiple comparisons via false discovery estimation [55], $p < 0.05$) in the obj-vis, subj-vis, and subj-inv condition, but not in the obj-inv condition (**Fig 3A**). In the three former visibility conditions, clusters with significant category information were located in bilateral fusiform gyrus, lingual gyrus, parahippocampal gyrus, and inferior occipital gyrus—areas overlapping with the high-level visual cortical areas defined in our ROI analyses. Compared to subjectively invisible stimuli, subjectively visible stimuli were associated with greater category information in the very same regions, and in bilateral fusiform gyrus in particular, thus confirming the results from the ROI analyses (**Fig 3B**). Additional clusters for subjectively visible stimuli were located in bilateral inferior temporal gyrus, mid-occipital gyrus, superior occipital gyrus, left superior parietal lobule, precuneus, and left amygdala. Compared to this effect of subjective visibility, the contrast of objectively visible with objectively invisible stimuli revealed greater and more widespread differences in category information in overlapping occipitotemporal regions, as well as additional effects in inferior frontal regions. When directly contrasting subjectively invisible with objectively invisible stimuli, there was greater category information for subjectively invisible stimuli in right fusiform gyrus and in left anterior fusiform gyrus extending into left inferior temporal gyrus (**Fig 3B**).

## Correlates of subjective awareness

As mask contrast covaried with subjective visibility, stronger category representations in the subj-vis than in the subj-inv condition could reflect differences in both stimulus strength and subjective awareness. To control for the effect of stimulus strength, we modeled brain responses in the subjective condition (separately for faces and houses) with a parametric regressor reflecting trial-by-trial mask contrast and an additional regressor for visibility (subj-vis, subj-inv). The visibility regressor was orthogonalized with respect to the mask-contrast regressor, assigning the mask-contrast regressor all shared variance, such that only variance not explained by the mask-contrast regressor was assigned to the visibility regressor [56]. Note that if fluctuations in subjective awareness and in mask contrast reflected partially shared processes, such as fluctuations in arousal or attention, these would be captured by the mask-contrast regressor, rendering this analysis a conservative approach for determining the neural correlates of subjective awareness.

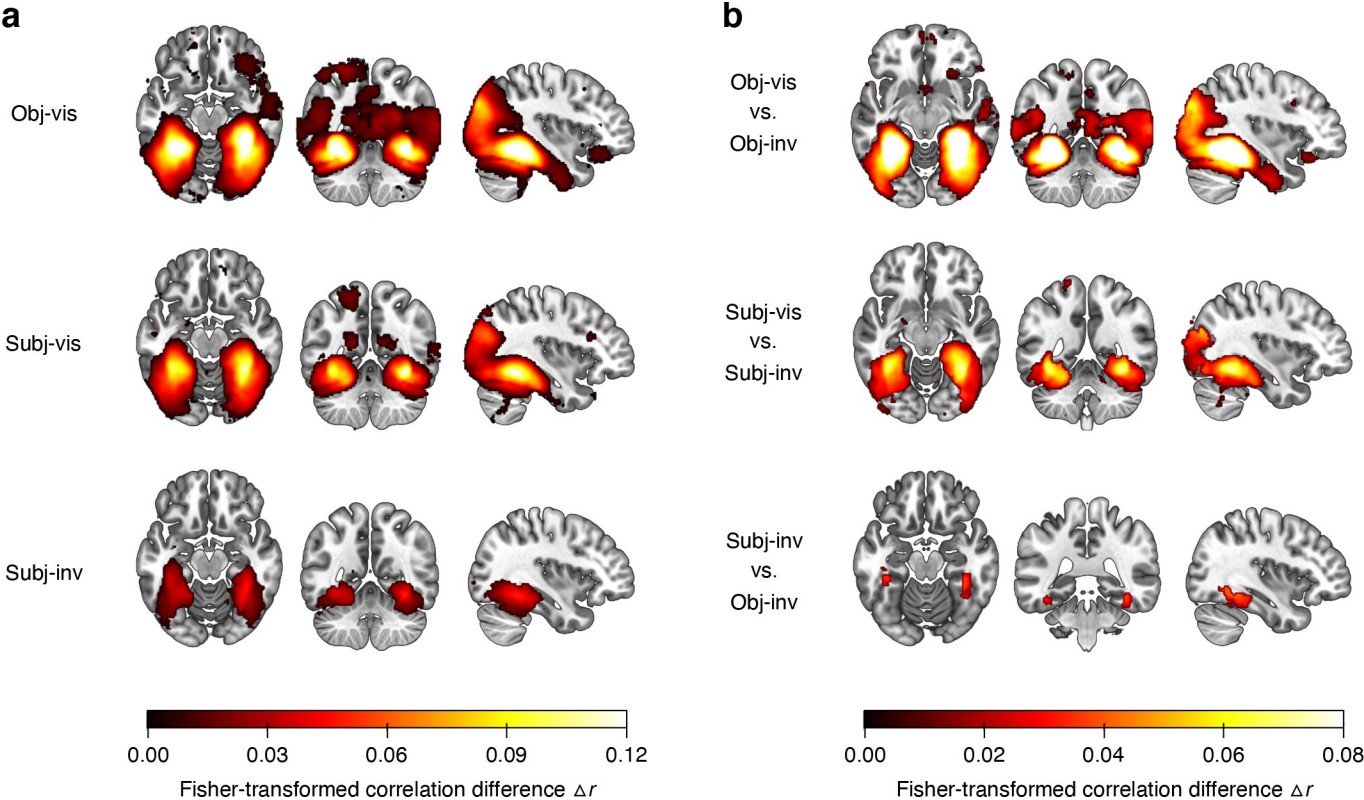

**Fig 3. (a)** Searchlight results showing clusters with significant category information across the whole brain, separately for the different visibility conditions. Only voxels surviving multiple comparison correction via false discovery estimation [55] ($p < 0.05$ FDR corrected) are shown. For the obj-inv condition, no clusters survived this statistical threshold. **(b)** Searchlight results showing the effect of subjective visibility by comparing the obj-vis to the obj-inv condition, the subj-vis to the subj-inv condition, and the effect of method for establishing invisibility by comparing the subj-inv to the obj-inv condition. Data underlying this figure are available on OSF (https://osf.io/qus5v/). FDR, false discovery rate; obj-inv, objectively invisible; obj-vis, objectively visible; subj-inv, subjectively invisible; subj-vis, subjectively visible.

Multivoxel pattern correlations between the localizer data and the face-/house patterns adjusted for mask contrast (**Fig 4A**) revealed strong effects of subjective visibility (one-sample $t$ test, two-tailed) in the more anterior areas LOC ($t_{(42)} = 3.57$, $p < 0.001$, $d_z = 0.54$, $BF_{10} = 32.03$) and FFA/PPA ($t_{(42)} = 3.08$, $p = 0.004$, $d_z = 0.47$, $BF_{10} = 9.45$), but only inconsistent effects in the more posterior areas V1 ($t_{(42)} = 1.96$, $p = 0.056$, $d_z = 0.30$, $BF_{10} = 0.94$) and OFA/OPA ($t_{(42)} = 2.21$, $p = 0.033$, $d_z = 0.34$, $BF_{10} = 1.46$). Note, however, that there were no significant differences between posterior and anterior ROIs (both $t_{(42)} < 1.37$, $p > 0.17$, $d_z < 0.21$, $BF_{01} > 2.57$), limiting claims about the spatial specificity of the effect. To determine whether subjective awareness enhanced category information specifically in occipitotemporal regions, we conducted a whole-brain searchlight analysis using the model controlling for mask contrast. Even at a more liberal statistical threshold ($p < 0.001$, uncorrected) this analysis revealed larger clusters with greater category information in subj-vis than in subj-inv only in bilateral fusiform gyrus (**Fig 4B**) and in bilateral mid-occipital gyrus (additional small clusters were located in superior parietal lobule). This further highlights the spatial specificity of the effect of subjective awareness.

## Correlates of blindsight-like discrimination performance

To test for brain areas involved in blindsight-like unconscious perception [25], we harnessed the fact that there was considerable interindividual variability in behavioral discriminability of

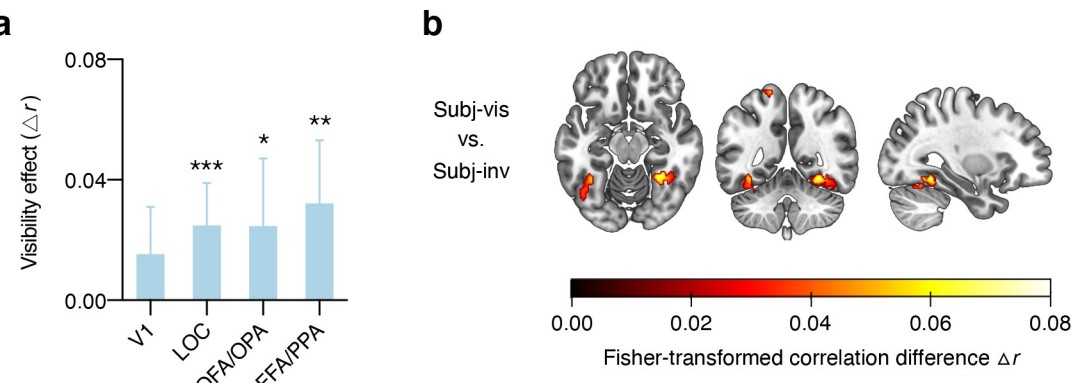

**Fig 4. Effect of subjective awareness, adjusted for differences in mask contrast between the subj-vis and subj-inv condition.** (a) Difference in multivoxel pattern correlations ($\Delta r$) between the subj-vis and subj-inv condition in the four ROIs. $^*p < 0.05$ $^{**}p < 0.01$ $^{***}p < 0.001$. (b) Searchlight results showing clusters with significantly greater category information in the subj-vis than in the subj-inv condition across the whole brain, controlling for mask contrast. Note that this map is thresholded at $p < 0.001$, uncorrected. Data underlying this figure are available on OSF (https://osf.io/qus5v/). FFA, fusiform face area; LOC, lateral occipital complex; OFA, occipital face area; OPA, occipital place area; PPA, parahippocampal place area; ROI, region of interest; subj-inv, subjectively invisible; subj-vis, subjectively visible; V1, primary visual cortex.

subjectively invisible stimuli (**Fig 5A**). The phenomenon of blindsight refers to above-chance behavioral performance in the absence of subjective awareness of the stimulus on which these behavioral effects are based [57]. For the subj-inv condition, correlations between perceptual discriminability (*d'*, behavioral performance) and category information in ROIs ($\Delta r$) were significant in LOC ($p < 0.001$), OFA/OPA ($p = 0.028$) and FFA/PPA ($p = 0.001$), but not in V1 ($p = 0.88$). One concern with these analyses is that mask contrast in the subj-inv condition was adjusted dynamically for each observer and thus differed between participants. Indeed, mask contrast was (negatively) correlated with behavioral performance ($r_{(41)} = -0.40$, $p = 0.008$, BF$_{10}$ = 5.74). We therefore computed partial correlations between perceptual discriminability and the ROI data, controlling for the effect of mask contrast. As can be seen in **Fig 5B**, also with this analysis there were significant correlations between discrimination performance and

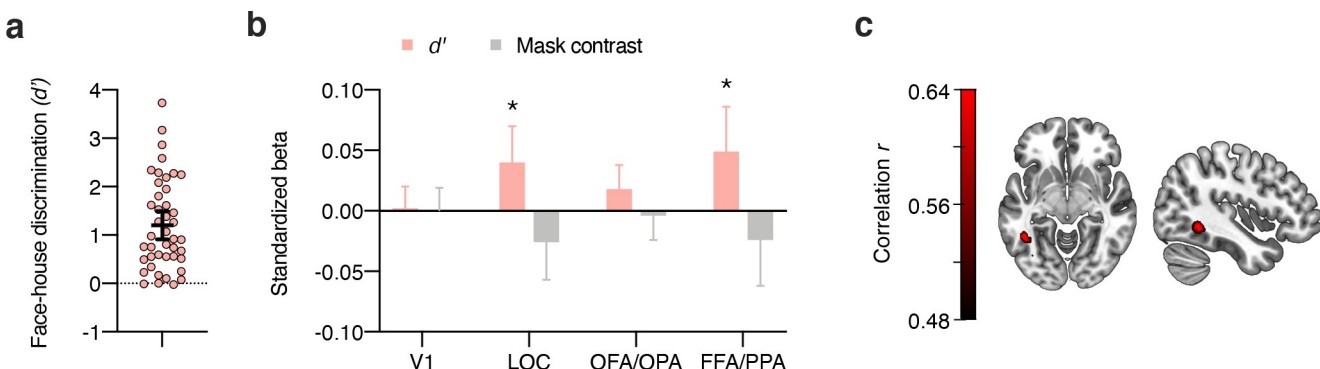

**Fig 5. Blindsight-like discrimination of subjectively invisible stimuli.** (a) Perceptual discrimination of subjectively invisible stimuli *(d')* was highly variable between individuals. Every circle represents a participant; the horizontal bar the mean, and the error bar the 95% confidence interval. (b) Multiple regression analyses of category information ($\Delta r$) in the four ROIs for the subj-inv condition, showing standardized beta weights for the predictors perceptual discrimination of subjectively invisible stimuli *(d')* and mask contrast in the subj-inv condition. Error bars show 95% confidence intervals. $^*p < 0.05$ $^{**}p < 0.01$. (c) Partial correlation between perceptual discrimination and the subj-inv whole-brain searchlight, adding the effect of mask contrast to the null model. Note that this map is thresholded at $p < 0.001$, uncorrected. Data underlying this figure are available on OSF (https://osf.io/qus5v/). FFA, fusiform face area; LOC, lateral occipital complex; OFA, occipital face area; OPA, occipital place area; area; PPA, parahippocampal place area; ROI, region of interest; subj-inv, subjectively invisible; V1, primary visual cortex.

category information in LOC ($r_{(41)} = 0.38$, $p = 0.013$, $BF_{10} = 4.68$) and in FFA/PPA, ($r_{(41)} = 0.38$, $p = 0.012$, $BF_{10} = 4.89$), but the correlation with OFA/OPA was not significant anymore ($r_{(41)} = 0.29$, $p = 0.064$, $BF_{10} = 1.43$). To examine the spatial specificity of these correlations, we calculated a correlation between perceptual discriminability and the subj-inv whole-brain searchlight. Even at a liberal statistical threshold ($p < 0.001$, uncorrected, **Fig 5C**), this analysis only revealed a cluster in left fusiform gyrus, further indicating the spatial specificity of the results.

## Discussion

In the absence of a gold standard for measuring consciousness, scientists are firmly divided into those who use subjective measures based on participant's reported experience and those who use objective measures based on participant's perceptual discrimination performance [11,16]. Using fMRI, we found that both objective and subjective visibility were associated with a clear posterior-to-anterior gradient: While there was little effect of visibility on category information in early visual cortex, higher-level visual cortex was strongly modulated by visibility. In frontal cortex, we did not obtain evidence for such effects of visibility on category representations. When investigating neural processing of invisible stimuli, subjective measures allow for greater stimulus strength (e.g., lower mask contrast) than objective measures, and this may result in greater estimates of unconscious processing. We found significant category information even for objectively invisible stimuli in both early, lower-level visual areas and in higher-level visual cortex, although representations in anterior, category-selective ventrotemporal areas (FFA/PPA) were less robust. Subjectively invisible stimuli were more strongly represented in object-selective visual cortex than objectively invisible stimuli. Furthermore, for subjectively invisible stimuli, similar to visible stimuli, there was a clear posterior-to-anterior gradient in visual cortex, with stronger category information in ventrotemporal cortex (FFA/PPA) than in occipital cortex (V1, OFA/OPA). For objectively invisible stimuli, category information remained virtually unchanged from early visual cortex to object- and category-selective areas. We interpret this as an indication that representations of objectively invisible stimuli are limited to visual (shape-related) object properties processed in early, lower-level visual areas, while subjectively invisible stimuli are processed up to more categorical levels of representation in higher-level category-selective areas. Although differences in multivoxel pattern correlations could reflect differences in response amplitude or changes in activation patterns, we obtained almost identical results in a univariate ROI analysis (see **S1 Text** and **S3 Fig**), and thus consider differences in response amplitude between face- and house-selective regions the more likely account for differences between visibility conditions. These differences in neural processing of subjectively versus objectively invisible stimuli can account for some of the current controversies on the scope and extent of unconscious processing [9,14].

Unconscious processing of objectively invisible stimuli has been notoriously difficult to demonstrate. Rendering stimuli perceptually indiscriminable through visual masking requires very low stimulus strength (e.g., high mask contrast), which strongly reduces neural responses in visual cortex, especially at higher levels in the cortical hierarchy [5]. Furthermore, above-chance performance in objective measures may be driven by unconscious processing [43]. As a result, the use of objective measures may underestimate unconscious processing. Indeed, several previous fMRI studies failed to obtain visual cortex responses to invisible faces [38,58], shapes [59], and objects [25,60]. Here, although maximum mask contrast was required for objective invisibility, perceptually indiscriminable faces and houses still evoked distinct activity patterns in visual cortex. Category information could be decoded from early visual cortex, from object-selective LOC, from category-selective areas in lateral occipital cortex (OFA/

OPA), and, less robustly, from ventrotemporal areas (FFA/PPA), while participants were unable to discriminate these categories, thereby satisfying the most rigorous criteria for establishing absence of awareness (see also **S1 Text** and **S2 Fig** for more details on this issue). Previous fMRI studies that found visual cortex responses to masked stimuli did not convincingly rule out conscious processing because these studies had very small sample sizes [33–38] and measured awareness with fewer trials [35,37,39–42] (or even fewer participants [38,40]), often in a separate block outside the scanner [35–41]. With low power, a nonsignificant effect in the awareness measure ($p > 0.05$) is not surprising and hard to interpret. Even if both the awareness measure and the fMRI data reflected the same underlying (conscious) process, failure to reject the null hypothesis is expected to occur more frequently for the measure with lower power (the awareness measure). To address these concerns, we tested a much larger sample than previous fMRI studies on unconscious processing, collected objective awareness measures on all 200 trials of the objectively invisible condition during scanning, and calculated BF to quantify the evidence favoring the null hypothesis of zero face/house discriminability versus the alternative hypothesis of above-chance discrimination performance [61]. Finally, no participants or trials needed to be excluded based on high awareness scores. Such data exclusion is a common procedure that risks severely inflating estimates of unconscious processing due to regression to the mean [62]. On a cautionary note, however, although we adopted the highest standards for demonstrating unconscious processing with fMRI to date, our claim of complete absence of awareness depends on specific statistical assumptions such as the prior in Bayesian analyses (also see **S1 Text** and **S2 Fig**). Notwithstanding these limitations, our findings provide more conclusive fMRI evidence for unconscious processing of objectively invisible stimuli in human visual cortex.

While representations of objectively invisible stimuli were of similar strength along the posterior–anterior axis from early visual cortex to object- and category-selective areas, subjectively invisible stimuli—similar to visible stimuli—were most strongly represented in category-selective areas in ventrotemporal cortex (FFA/PPA). These differences in neural representation may provide an account for discrepant findings on the influence of unconscious stimuli on behavior, for example in visual priming experiments [13,63,64]. For objectively invisible stimuli, our findings predict priming effects based on visual features such as shape, but not based on semantic meaning such as category membership (e.g., gender, emotion, or animacy). Indeed, while response priming from objectively invisible shapes (e.g., left- versus right-pointing arrow [65] or square versus diamond [66]) is well established, semantic priming from objectively invisible pictures, where visual effects are ruled out (e.g., a picture of an animal priming a word referring to an animal), is heavily debated [13,63,64]. When objective discrimination performance is well above chance, as for subjectively invisible stimuli in the present study, masked primes elicit robust semantic processing [67]. Recent studies indicate that subjectively invisible stimuli reach even higher levels of processing, including crossmodal semantic integration [68] and working memory [69]. The present findings will resonate with the idea that subjectively invisible stimuli are processed in a way that is qualitatively similar to (clearly) visible stimuli [14].

Activity patterns in LOC and FFA/PPA also predicted perceptual discrimination performance for subjectively invisible stimuli, representing a neural correlate of blindsight-like unconscious perception in healthy human observers [25]. For proponents of subjective measures to study unconscious processes, this could be regarded as strong evidence that unconscious perception shares neural mechanisms with conscious perception, providing support for the idea that consciousness has little functional role in human perception and cognition [14]. However, for proponents of objective awareness measures, above-chance performance for subjectively invisible stimuli simply means that these stimuli were in fact not invisible. According

to this view, differences in perceptual performance on subjectively invisible trials could reflect differences in response criteria (willingness to say "visible" to barely visible stimuli [26,29–32]). In our staircasing procedure, mask contrast was continuously adjusted based on subjective visibility, such that a more liberal criterion (greater willingness to say "visible") would result in higher average mask contrast and this could account for lower performance. Patterns in LOC and FFA/PPA were better predicted by perceptual performance than by mask contrast, and perceptual performance explained a significant portion of the neural variance taking into account the variance explained by mask strength. However, without a measure of sensitivity and criterion for the visibility task (e.g., by including stimulus-absent trials), our results cannot rule out criterion effects.

Because it is the very nature of subjective measures that visibility judgments are subject to the observer's own interpretation and response criterion, it is possible that results would have differed for the subjectively invisible condition had we adopted another visibility scale. Although binary visibility scales such as the one used here have been widely adopted [26], for example in studies using masking [70,71], attentional blink [72], dichoptic fusion [73], and interocular suppression [74,75]; recently, the four-point perceptual awareness scale (PAS) [17] has gained in popularity. Depending on specific instructions and observer's interpretations, the present "invisible" rating may have corresponded to the lowest level or to the two lowest PAS levels. While some PAS studies defined invisibility as the two lowest PAS levels [68,76], using only the lowest level is more common in the literature [20,21,23]. Compared to this approach, our subjective measure may have been more liberal in defining invisibility, resulting in greater estimates of blindsight-like above-chance performance for subjectively invisible stimuli. This highlights a general problem with subjective measures. Subjective states are "private", i.e., they cannot be externally falsified, so that differences between scales, observers, or studies cannot be unequivocally linked to differences in perceptual states. One solution to this problem is to adopt objective measures, which may come at the cost of failing to capture the subjective quality of conscious experience and which may be influenced by unconscious processes. For these reasons, it is difficult to determine whether above-chance performance for subjectively invisible stimuli reflects unconscious processing or a conservative response criterion in the subjective measure. One promising avenue for future studies is to adopt recently developed criterion-free measures of subjective awareness such as meta-$d'$ based on confidence ratings [77].

Nevertheless, by comparing stimuli of similar stimulus strength reported as "visible" versus "invisible," subjective measures have often been adopted to reveal the neural correlates of subjective awareness. Although the contrast of subjectively visible and invisible trials may be confounded with factors unrelated to awareness (e.g., alertness, attention), many previous fMRI studies adopted this approach, often using binocular rivalry or other bistable stimuli, and typically found stronger visual cortex responses for stimuli reported as visible [12]. Our finding of better pattern discrimination of subjectively visible stimuli in object- and category-selective visual cortex is consistent with the idea that subjective awareness is related to enhanced activity in the very same brain areas that are specialized for processing those stimuli [78,79]. Whether activity in this posterior occipito–temporo–parietal "hot zone" [80] is sufficient for awareness, or whether awareness requires additional activity in prefrontal cortex [7,81], is subject of ongoing debate. Recent studies using so-called "no-report" paradigms indicate that activity in frontal cortex may reflect post-perceptual processes related to reporting awareness rather than to content-specific perceptual awareness per se [82]. Although establishing absence of awareness online during imaging required trial-by-trial visibility and discrimination responses, our approach of correlating activity with "benchmark" patterns from an independent localizer scan where participants had a different task reduced the influence of post-perceptual effects

(also see **S1 Text**) [51,52]. Ruling out task-specific representations in this way, there was no evidence for prefrontal cortex involvement (see **S5 and S6 Figs** and **S1 Text** for evidence that activity in a frontoparietal network distinguishes visible versus invisible conditions, irrespective of stimulus category).

The marked differences in neural representation of subjectively versus objectively invisible stimuli revealed here imply that models where "unconscious processing" is seen as a unitary mechanism or concept, without further specifying how absence of awareness was established, will not provide a good fit to the full range of empirical data. Rendering stimuli subjectively or objectively invisible results in different estimates of unconscious information processing in the human brain. We provide the first conclusive evidence that even fully indiscriminable, objectively invisible stimuli can be decoded from patterns of fMRI activity in human visual cortex, albeit at a drastically lower strength than visible stimuli. Previous studies that made such claims [33–38] did not convincingly establish absence of awareness during scanning and/or had very low statistical power (small sample sizes), resulting in high probability of false positives in the fMRI data [83] and of false negatives in the awareness measure [84,85]. However, only subjectively invisible, but not objectively invisible, stimuli were increasingly processed along a posterior–anterior gradient, with greater category information in category-selective ventrotemporal cortex than in occipital cortex.

## Methods

### Participants

The experiments were approved by the University of Amsterdam Ethics Committee (approval numbers 2019-BC-10091 and 2019-BC-10347), and all procedures were conducted according to the principles expressed in the Declaration of Helsinki. Volunteers were recruited from the University of Amsterdam participant pool. Participants were mostly students who received either course credit or a monetary compensation for their participation. All participants reported normal or corrected-to-normal vision, were naïve to the research question, and provided written informed consent. In the fMRI experiment, we scanned 54 participants. This sample size was constrained by the resources provided by an fMRI grant awarded to the first author. We tested as many participants as possible within the allotted scanning time. Eleven participants were excluded from all data analyses: three were excluded because they did not finish the experiment, two because their anatomical scans were corrupted, and six because of failure to follow the instructions inside the scanner (very few button presses, holding down the buttons continuously, or reporting stimulus visibility in almost all trials of the obj-inv condition). The final sample consisted of 43 participants (24 female, mean age 22.7 years, SD 3.9, range 18 to 37 years). In the behavioral masking efficiency experiment, there were 18 participants, one of which was excluded due to a coding error (unbalanced experimental conditions), resulting in a final sample of 17 participants (12 female, mean age 21.8 years, SD 4.7, range 18 to 31 years).

### Stimuli

In the scanner, stimuli were presented on a 32-inch LCD screen for MRI (1920 × 1080 pixel resolution, 120-Hz refresh rate) seen from a viewing distance of approximately 150 cm through a mirror mounted on the head coil. The experiment was programmed in MATLAB using the Psychtoolbox [86] functions. Stimulus presentation was synchronized with the 8.3-ms vertical refresh cycle of the screen. Stimuli were ten face photographs of neutral expression from the FACES database [87] and ten house photographs (taken from the front) selected from the internet. This selection was based on informal pilot testing with a larger set of

stimulus exemplars. Pilot results indicated that the selected exemplars were best matched on subjective visibility. Photographs were cropped to an oval of 240 × 336 pixels containing only the inner features of faces and houses, converted to grayscale, and the oval was assigned identical mean luminance and contrast (in RGB values M = 127.5, SD = 17.5). The remainder of the stimulus rectangle as well as the screen background were mid-gray (RGB value 127.5). Masks filled the whole stimulus rectangle and consisted of a randomly generated arrangement of overlapping rectangles and—in a lesser number—circles in various sizes and levels of gray. We created a set of 100 masks, from which one forward mask and one backward mask were randomly selected on every trial. The contrast of these masks differed between visibility conditions and varied between 2% and 100%. Throughout the experiment, all stimuli and masks were presented within an 8-pixel wide light gray frame (248 × 344 pixels, RGB value 143).

In the obj-vis condition, mask contrast was set to 2% to achieve clear visibility of the face/house stimuli. In the obj-inv condition, mask contrast was set to 100%, such that participants' ability to discriminate between faces and houses was expected not to differ significantly different from chance (based on the results from the masking efficiency experiment). In the subjective condition, mask contrast was adjusted through an adaptive 1-up 1-down staircase procedure: On the first trial of each run, mask contrast started at 18%; following a "visible" response, mask contrast on the next trial was increased by 4%; following an "invisible" response, mask contrast was lowered by 4% (minimum contrast 2%, maximum contrast 100%). This adjustment was intended to yield a roughly similar number of subj-vis trials and subj-inv trials.

## Procedure

On every trial, a forward- and backward-masked face or house stimulus was presented, and participants indicated stimulus category and visibility. Every trial began with 292 ms of fixation on, followed by 392 ms fixation off, 92 ms forward mask, 16.7 ms face/house stimulus (two screen refresh cycles), and 192 ms backward mask. Next, a response screen prompted participants to use one of four buttons to indicate stimulus category and visibility, using their left hand for the two left buttons and their right hand for the two right buttons (using a button box inside the scanner and a standard keyboard outside the scanner). The left-most button represented "face, visible," and the other button on the left represented "face, invisible." The right-most button represented "house, visible," and the other button on the right "house, invisible." Pilot experiments had shown that this compound response and response mapping was intuitive, and all participants received extensive training in using the buttons. There was a response window of 1.8 s in which participants could enter their response. The trial ended with a fixation period of variable (jittered) duration (selected from a uniform distribution between 100 and 900 ms with as many values as trials per fMRI run or as trials per behavioral experiment). The interstimulus interval ranged between 2.9 and 3.7 s (while brain activity was measured with a fast echo-planar imaging [EPI] sequence with a repetition time of 1.6 s; see below).

## Instructions

Before beginning the experiment, participants received detailed written and verbal instructions. They were informed that they would be presented with pictures of faces and houses and that masking would be used to degrade visibility of these stimuli, such that some of the stimuli would be visible and some would be invisible. They were instructed to indicate both stimulus category and visibility using the compound response. Participants were informed that when they had absolutely no idea of what category the stimulus represented (i.e., if they did not see anything that indicated that the picture was a face or a house), they should indicate "invisible"

and take a guess. If they had some (vague) idea of what the stimulus category could be, they should indicate "visible." Participants were informed about the 1.8-s response window, but instructions emphasized that (after practice) this would be more than enough time to provide an accurate response, such that there was no speed pressure and that responses should be as accurate as possible.

Before entering the scanner, participants completed one practice run at a computer outside the scanner. At the beginning of the fMRI session, an anatomical scan was acquired, and participants completed another practice run, which was followed by five runs of the main experiment. fMRI runs started and ended with 6.4 s of fixation. In the following 160 trials of a run, there were 40 trials of the obj-vis condition, 40 trials of the obj-inv condition, and 80 trials of the subjective condition (here, the number of trials in the subj-vis and subj-inv condition depended on the participant's response). Within each condition, each combination of two stimulus categories (faces and houses) and ten stimulus exemplars occurred equally often. Trial order was randomized. In total, there were 200 trials of the obj-vis condition, 200 trials of the obj-inv condition, and 400 trials of the subjective condition.

### Localizer run

At the end of the fMRI session, we acquired a functional localizer scan to localize face- and house-responsive voxels in visual cortex. In the localizer, the same stimulus exemplars as in the main experiment were presented in a standard design for localizing category-selective brain areas: Faces and houses were displayed in separate 16-s blocks, where a series of 16 faces or houses was presented unmasked for 750 ms each, followed by 250-ms fixation. There were 20 face blocks and 20 house blocks in alternating order. After every four blocks, there were 16 s of fixation. Within a block, stimulus exemplars were randomly ordered with the constraint that the same exemplar could only be presented two times per block. Participants were instructed to press a button when there was a repetition of an exemplar (we did not record these button presses).

### fMRI acquisition

MRI data were collected using a 3 Tesla Philips Achieva MRI scanner (Philips, Eindhoven, the Netherlands) with a 32-channel head coil. At the beginning of the fMRI session, an anatomical scan was acquired using a T1-weighted gradient-echo sequence (220 slices, 1-mm isotropic voxels). Functional images were acquired using a T2*-weighted multiband EPI sequence (56 slices, flip angle 70˚, TR 1600 ms, TE 30 ms, 2-mm isotropic voxels). During the main experiment, 348 volumes were recorded per run (for a total of five runs lasting 9:30 min each), and 248 volumes during the localizer run (lasting 6:30 min).

### Behavioral masking efficiency experiment

The masking efficiency experiment was conducted to determine the required masking settings to achieve objective invisibility, i.e., a masking setting at which participants' ability to discriminate between faces and houses was not significantly above chance. Stimuli were presented on a 24-inch LCD screen (1920 × 1080 pixel resolution, 120-Hz refresh rate) seen from a free viewing distance of approximately 80 cm. The masking efficiency experiment was identical to the fMRI experiment, except that nine fixed masking strengths ($log_{10}$-scale between 2% and 100% contrast) were used. There were 540 trials, in which each combination of two stimulus categories (faces and houses), ten stimulus exemplars, and nine masking strengths (2, 3.3, 5.3, 8.7, 14.1, 23.1, 37.6, 61.3, 100%) occurred equally often. Trial order was randomized, and there were three obligatory breaks.

## Analyses

### Behavioral data

Trials with no response (on average less than 3%) were excluded from analyses of behavior and fMRI. We calculated the signal detection theory (SDT) sensitivity index $d'$ as a measure of objective perceptual discriminability of faces and houses: "Face" responses were coded as hits in face trials and as false alarms in house trials. Hit and false alarm rates of 0 or 1 were converted to $1/(2N)$ and $1-1/(2N)$, respectively, with $N$ being the number of trials on which the rates were based [27]. Finally, the $z$-transformed false alarm rate was then subtracted from the $z$-transformed hit rate to yield $d'$.

### fMRI data preprocessing

Neuroimaging data were preprocessed using fMRIPrep 1.3.2 [88]. For the structural images, preprocessing steps included intensity non-uniformity correction, skull stripping, surface reconstruction, spatial normalization to the ICBM 2009c Nonlinear Asymmetrical template version 2009c [89] (with nonlinear registration), and brain tissue segmentation. Preprocessing steps for the functional images included susceptibility distortion correction, co-registration to the structural image, estimation of head motion parameters, resampling of the BOLD time series to the template, and high-pass filtering (using a discrete cosine filter with 128 s cut-off).

### fMRI data modeling

SPM12 was used to fit a general linear model (GLM) to the data from the localizer run and to the data for each run of the main experiment. For the localizer, the GLM contained two regressors (faces and houses) and six regressors of no interest (head motion parameters). For each run of the main experiment, the GLM contained eight regressors (four visibility conditions; objectively and subjectively visible and invisible, and two stimulus categories; faces and houses), as well as six motion regressors. Regressors were convolved with a standard hemodynamic response function, as included in SPM12. The resulting beta weights for each voxel were used as the data points for the following analyses.

### Multivoxel pattern analyses

Multivoxel pattern analysis (MVPA; using the CoSMoMVPA toolbox [90]) was used to identify brain regions carrying category-specific information in their activity patterns, i.e., activity that reliably distinguished between face and house stimuli. Activity patterns were correlated between the block-design localizer and the event-related main experiment. For every participant and for each of the four visibility conditions (obj-vis, subj-vis, subj-inv, and obj-inv), betas from the localizer for one stimulus category (faces and houses) were correlated with betas from the main experiment for the same stimulus category, yielding within-category correlations, and with betas from the main experiment for the other stimulus category, yielding between-category correlations. Correlations were then Fisher $z$-transformed, the two within-category correlations were averaged, and the two between-category correlations were averaged. Finally, between-category correlations were subtracted from within-category correlations, yielding a correlation difference $\Delta r$ for each visibility condition [51,52]. Positive correlation differences indicate that activity patterns carry information about stimulus category.

### Region of interest analyses

Pattern correlation differences were calculated for four different (bilateral) ROIs. A probabilistic atlas of retinotopic cortex [91] was used to define bilateral early visual cortex (V1). A

functional group atlas [92] was used to define bilateral LOC, posterior category-selective areas (face-selective OFA and scene-/house-selective OPA [transverse occipital sulcus]), and anterior category-selective areas (face-selective FFA and scene-/house-selective PPA). For every participant, the localizer data were used to select the 100 most face-responsive and the 100 most house-responsive voxels within each of these ROIs (based on a *t* test comparing face and house responses; for similar results with other ROI definitions, see **S4 Fig** and **S1 Text**).

### Searchlight analyses

We additionally ran searchlight analyses [54] to identify activity patterns that distinguished between categories across the whole brain. For these searchlights, we repeatedly calculated pattern correlation differences in the same way as described for the ROI analyses but for a moving sphere with a radius of five voxels (524 voxels in each sphere) which was centered on every voxel in the functional images of every participant (for searchlight analyses based on the data from the main experiment only, see **S1 Text** and **S5 and S6 Figs**).

### Controlling for mask contrast in the subjective condition

To control for trial-by-trial differences in mask contrast in the subjective condition, another GLM including a parametric regressor for mask contrast and an additional regressor for visibility, separately for faces and houses, was fit to the data from the subjective condition (all other aspects of the GLM were the same as described above). The visibility regressor was orthogonalized with respect to the mask-contrast regressor, such that the visibility regressor was assigned only the variance not explained by the mask-contrast regressor. Both the mask-contrast and the visibility regressor were mean-centered (run-wise, separately for faces and houses).

### Statistics

For behavior and ROI data, we report both standard frequentist statistics and BFs calculated in JASP [93] with default prior scales (Cauchy distribution, scale 0.707). When frequentist statistics indicate a significant effect, the corresponding BF is reported as a quantification of the evidence for the alternative hypothesis ($BF_{10}$); when the effect is not significant, the reported BF quantifies the evidence for the null hypothesis ($BF_{01}$). To demonstrate absence of awareness in our objective measure, the directional $BF_{0+}$ quantifies evidence for null sensitivity compared to the alternative of above-chance performance. For multifactorial ANOVAs, we report the inclusion BF quantifying the evidence for all models containing a particular effect compared to all models without that effect. To test for category information in ROIs, the directional $BF_{+0}$ quantified evidence for above-chance information compared to the alternative of zero information. For the searchlight group maps, results were corrected for multiple comparisons via false discovery rate (FDR) corrections [55] ($p < 0.05$).

### Supporting information

**S1 Text. Additional results from the masking efficiency and the fMRI experiment, including evidence for objective invisibility, univariate ROI results, results for different ROI definitions, and main experiment searchlight results.** fMRI, functional magnetic resonance imaging; ROI, region of interest.
(PDF)

**S1 Table. Reliability estimates of face-house discrimination performance in the behavioral masking efficiency experiment.** The diagonal (in bold) shows within-condition reliability

estimates, which were calculated by repeatedly correlating performance from 2 randomly determined halves of the data set. The other cells show correlations between performance from different mask conditions.
(PDF)

**S1 Fig. Additional results from the behavioral masking efficiency experiment. (a)** Normalized results. For every level of mask contrast mean face-house discriminability and mean subjective visibility were scaled between 0 and 1, and a logistic function was fit to the resulting normalized scores. **(b)** Face-house discriminability for individual participants at the three highest mask contrasts (where mean *d'* was below 1) in the masking efficiency experiment. Data underlying this figure are available on OSF (https://osf.io/qus5v/).
(TIF)

**S2 Fig. Additional behavioral results from the fMRI experiment. (a)** Individual participant's face-house discrimination performance in the obj-inv condition as proportion correct. Every gray circle represents a participant, and the red circle shows the group mean with its 95% confidence interval. **(b)** Histogram showing the cumulative distribution of *p*-values from the 1-sided binomial tests of face-house discrimination accuracy in the obj-inv condition. Data underlying this figure are available on OSF (https://osf.io/qus5v/). fMRI, functional magnetic resonance imaging; obj-inv, objectively invisible.
(TIF)

**S3 Fig. Univariate fMRI results.** Responses to a voxel's preferred category vs. its non-preferred category (as determined by the independent localizer) for the four visibility conditions in the main experiment, averaged across the voxels in each brain region. Bars show the mean beta difference between preferred and non-preferred categories ($\Delta\beta$). Error bars represent 95% confidence intervals. **(a)** Results for early visual cortex (V1) vs. object-selective visual cortex (LOC). **(b)** Results for posterior category-selective areas (OFA/OPA) vs. anterior category-selective areas (FFA/PPA). Data underlying this figure are available on OSF (https://osf.io/qus5v/). FFA, fusiform face area; fMRI, functional magnetic resonance imaging; LOC, lateral occipital complex; OFA, occipital face area; OPA, occipital place area; PPA, parahippocampal place area; V1, primary visual cortex.
(TIFF)

**S4 Fig. Category information as a function of different ROI definitions.** Multivoxel pattern correlations between the four visibility conditions in the main experiment and the independent localizer, for a range of different ROI definitions (containing the 10–500 most face- and house-responsive voxels). Symbols show the mean difference between within-category and between-category multivoxel pattern correlations ($\Delta r$). For better readability, error bars represent SEMs. Arrows indicate the ROI definition adopted for the results presented in the main paper. **(a)** Results for early visual cortex (V1, left panel) vs. object-selective visual cortex (LOC, right panel). **(b)** Results for posterior category-selective areas (OFA/OPA, left panel) vs. anterior category-selective areas (FFA/PPA, right panel). Data underlying this figure are available on OSF (https://osf.io/qus5v/). FFA, fusiform face area; LOC, lateral occipital complex; OFA, occipital face area; OPA, occipital place area; PPA, parahippocampal place area; ROI, region of interest; V1, primary visual cortex.
(TIFF)

**S5 Fig. Main experiment searchlight: category information. (a)** Results from the additional searchlight analyses of the main experiment only, showing clusters with significant category information across the whole brain, separately for the different visibility conditions. Slices

were selected to highlight motor cortex. Only voxels surviving multiple comparison correction via false discovery estimation ($p < 0.05$) are shown. For the objectively invisible condition, no clusters survived this statistical threshold. **(b)** Searchlight results showing the effect of subjective visibility by comparing the subj-vis to the subj-inv condition and the effect of method for establishing invisibility by comparing the subj-inv to the obj-inv condition. Data underlying this figure are available on OSF (https://osf.io/qus5v/). obj-inv, objectively invisible; subj-inv, subjectively invisible; subj-vis, subjectively visible.
(TIF)

**S6 Fig. Main experiment searchlight: visibility information.** Results from searchlight analyses of the main experiment only, showing clusters with significant information about stimulus visibility across the whole brain (independent of stimulus category), separately for the objective condition (comparing obj-vis to obj-inv) and for the subjective condition (comparing subj-vis to subj-inv). Slices show the right hemisphere and were selected to highlight ventro-temporal regions, parietal cortex, and inferior frontal gyrus in both conditions. Only voxels surviving multiple comparison correction via false discovery estimation ($p < 0.05$) are shown. Data underlying this figure are available on OSF (https://osf.io/qus5v/). obj-inv, objectively invisible; obj-vis, objectively visible; subj-inv, subjectively invisible; subj-vis, subjectively visible.
(TIF)

## Acknowledgments

We thank T. Beemsterboer, T. Derckx, and S. Rasche for help with data collection and L. Snoek for implementing the standardized "fMRIPrep" pipeline.

## Author Contributions

**Conceptualization:** Timo Stein, Daniel Kaiser, Johannes J. Fahrenfort, Simon van Gaal.

**Formal analysis:** Timo Stein, Daniel Kaiser.

**Funding acquisition:** Timo Stein, Simon van Gaal.

**Investigation:** Timo Stein.

**Methodology:** Timo Stein, Daniel Kaiser, Johannes J. Fahrenfort, Simon van Gaal.

**Project administration:** Timo Stein.

**Resources:** Timo Stein.

**Software:** Daniel Kaiser.

**Supervision:** Timo Stein.

**Writing – original draft:** Timo Stein, Daniel Kaiser, Simon van Gaal.

**Writing – review & editing:** Timo Stein, Daniel Kaiser, Johannes J. Fahrenfort, Simon van Gaal.

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
