## [Editor Report · Decision Letter 0]

25 Nov 2020

Dear Dr Stein, 

Thank you for submitting your manuscript entitled "Processing of subjectively and objectively invisible stimuli in human visual cortex" for consideration as a Research Article by PLOS Biology.

Your manuscript has now been evaluated by the PLOS Biology editorial staff, as well as by an academic editor with relevant expertise, and I am writing to let you know that we would like to send your submission out for external peer review.

Please re-submit your manuscript within two working days, i.e. by Dec 01 2020 11:59PM. (Note the office will be close Thursday 26 and Friday 27 of November). 

Kind regards,

Gabriel Gasque, Ph.D.,

Senior Editor

PLOS Biology

---

## [Decision Letter · Decision Letter 1]

3 Feb 2021

Dear Dr Stein,

Thank you very much for submitting your manuscript "Processing of subjectively and objectively invisible stimuli in human visual cortex" for consideration as a Research Article at PLOS Biology. Your manuscript has been evaluated by the PLOS Biology editors, by an academic editor with relevant expertise, and by two independent reviewers. Please accept my apologies for the delay in sending the decision below to you.

In light of the reviews and the academic editor's detailed comments (below), we are pleased to offer you the opportunity to address the comments from the academic editor and from the reviewers in a revised version that we anticipate should not take you very long. We will then assess your revised manuscript and your response to the academic editor and reviewers' comments and we may consult the reviewers again.

We expect to receive your revised manuscript within 1 month.

**IMPORTANT - SUBMITTING YOUR REVISION**

Your revisions should address the specific points made by the academic editor and by each reviewer. As you will see from their detailed comments, the academic editor and the two reviewers feel that this topic is important and timely, and note the careful study design. There was, however, some concerns (raised most strongly by reviewer 3) with the way in which objective and subjective invisibility was defined and achieved - that reviewers argue could affect the study interpretations. Please address these comments thoroughly.

Please also make sure to address the data and other policy-related requests noted at the end of this email.

Please submit the following files along with your revised manuscript:

*Resubmission Checklist*

*Published Peer Review*

*PLOS Data Policy*

*Blot and Gel Data Policy*

Sincerely,

Gabriel Gasque, Ph.D.,

Senior Editor,

ggasque@plos.org,

PLOS Biology

ETHICS:

-- Please indicate within your manuscript if your experimental procedures adhered to principles expressed in the Declaration of Helsinki or any other national or international ethical guidelines.

DATA:

I note that you have stated your data in confidential. Please refer to these guidelines for accepted data access restrictions: https://journals.plos.org/plosbiology/s/data-availability#loc-acceptable-data-access-restrictions

Note, nonetheless, that we do not require the raw data. Rather, we ask for all individual quantitative observations that underlie the data summarized in the figures and results of your paper. For an example see here: http://www.plosbiology.org/article/info%3Adoi%2F10.1371%2Fjournal.pbio.1001908#s5

These data can be made available in one of the following forms:

Regardless of the method selected, please ensure that you provide the individual numerical values that underlie the summary data displayed in the following figure panels: Figures 1b-e, 2ab, 3ab, 4ab, 5a-c, S1bc, S2ab, S3ab, S4ab, and S5 (Please note that there are two figures S3, I thus renumbered S3 “(a) Results from the additional searchlight analyses of the main experiment only, …” as S4, and renumbered S4 as S5).

Please also ensure that each figure legend in your manuscript includes information on where the underlying data can be found and that your supplemental data file/s has/have a legend.

REVIEWS:

Academic Editor: The paper by Stein et al addresses a fundamental question in consciousness research- to what extent invisible stimuli nevertheless activate in an informative manner the human visual system. This is an important issue- since it bears critical importance to any theory that attempts to delineate the neuronal mechanisms that underlie conscious vs. non-conscious visual processes.

While this question has been addressed by a number of previous studies- these studies were notoriously plagued by ungrounded assumptions and poor methodology. Stein et al has made, to my mind, an important and decisive advance in this important field by carefully noting these previous problems and designing their study in a rigorous manner that impressively addresses these problems at their core. In that sense I feel that Stein et al bring a breath of fresh air to this problematic field- and provide conclusive and convincing findings in which the confounds and methodological mistakes are cleared out. I feel strongly that this work will provide an important and significant advance in this major field of human vision research.

Below are my comments following the flow of the manuscript itself.

Abstract:

"We show that neural representations of objectively invisible faces and houses

are limited to visual (shape-related) object properties, while subjectively invisible stimuli

are processed up to more abstract, categorical levels of representation"- I think the contrast between visual and abstract is misleading- it implies that category-selective areas are not visual- which is unfounded. The authors should use more neutral terms- such as early or lower-level vs. higher level visual areas.

Results:

Figure 1:

It will be informative if the authors show the fit between panels b and c- by normalizing the values between 0 and 100% and superimposing the two curves. The fit should be pointed out, as well as the highly non-linear (sigmoid) nature of the dependence on mask contrast. 

Discussion:

General points:

It is important that the authors discuss the meaning of the "stronger representation" to use their terminology when analyzed through multivariate correlation patterns. Specifically- a stronger vs weaker representation may result from at least two main sources: first, a weaker univariate activation- i.e. lower signal to noise responses in each voxel. Second- it may result from a change in the pattern of activation elicited during the weak vs strong responses-resulting in weaker and stronger similarity to the bench mark patterns. This point needs to be clarified- and kept in mind when interpreting the results. Just as an example is the following conclusion: "The present findings will resonate with the idea that subjectively invisible stimuli are processed in a way that is qualitatively similar to (clearly) visible stimuli"- In this statement the authors, presumably, assume that the patterns of activation for the subjectively invisible and visible stimuli are similar-differing only in signal amplitude. However this conclusion ignores the fact that the massive reduction in representation strength for subjectively invisible stimuli may be the result of a corresponding change in the pattern of activation making it dissimilar to the benchmark pattern. 

A second major methodological issue that should be discussed is the critical difference in the response properties to visible stimuli in early vs high order cortex- specifically in this case to the mask vs. target stimuli. It is quite likely that the weak category representations they find in V1 during the visible conditions is due to literally masking the category information by the much stronger mask response- since the fMRI method is too sluggish to temporally separate out the target response from the far larger response to the masks. This can easily lead to an erroneous underestimation of the low-level differences between the categories in V1 even for visible stimuli. This point should be addressed by the authors.

Specific points:

Before jumping into the discussion of the brain activations associated with the invisible conditions- it is important that the authors will first note the main, and very striking, effects demonstrated by their study. In particular-the very dramatic change in the "visibility effect"- i.e. the difference between visible vs invisible conditions- as one moves from early (essentially no difference) to high order visual cortex (massive difference). Similarly they should point in this context the abolishment of such visibility effect in frontal cortex. 

"Rendering stimuli perceptually indiscriminable requires very low stimulus strength" This is incorrect- there are cases where strong stimuli can be rendered invisible - such as binocular rivalry, crowding and motion-induced blindness.

"above-chance performance for subjectively invisible stimuli simply means that

these stimuli were in fact not invisible"- 

A possible analysis that may help clarify this conundrum could be to analyze separately those trials in which participants marked the target as invisible but guessed the category correctly vs. those subjectively invisible trials in which the participants guessed incorrectly. 

"objectively invisible stimuli can be decoded from patterns of fMRI activity in human visual cortex- to be fair to their own data the authors should add "albeit at a drastically lower strength than visible stimuli"

Reviewer #2: This is a very interesting and carefully conducted study. The methods and results are solid. I have no issues with the design or analyses. My only concerns are with the interpretation of the results and the way in which the conclusions and implications are framed in the abstract vs. discussion.

1. On page 14 (middle of first full paragraph), the authors state, "Category information could be decoded from early visual cortex, from object-selective LOC, from the (largely overlapping) category-selective areas in lateral occipital cortex (OFA/OPA), and, less robustly, from ventrotemporal areas (FFA/PPA), while participants were unable to discriminate these categories, thereby satisfying the most rigorous criteria for establishing absence of awareness"

I don't quite understand how this conclusion is consistent with the general point of the paper as outlined in the abstract. To render stimuli objectively invisible, a stronger mask was used, and this stronger mask led to weaker stimulus-evoked activity patterns in all of the brain regions analyzed. But, even with objective invisibility, the authors still find above chance decoding in all of these regions, so doesn't this mean that even with stricter criteria and improved methods (objective instead of subjective invisibility), the extent of unconscious visual processing is quite vast? Perhaps I'm not understanding the authors' interpretation of (A) the existence of significant multi-voxel patterns vs. (B) magnitude differences between subj vs. obj invisibility. (A) seems more relevant to the general question of unconscious information processing while (B) seems more relevant to a methodological nuance. Indeed, on the following page, the authors state, "Notwithstanding these limitations, our findings provide more conclusive fMRI evidence for unconscious processing of objectively invisible stimuli in human visual cortex."

My general concern here is that what is stated in the abstract doesn't really match up with this main conclusion. The abstract states, "These results demonstrate that the hypothesized extent of unconscious information processing is determined by the measurement approach. Furthermore, our data show that subjective and objective approaches are associated with different neural correlates of consciousness and thus have implications for neural theories of consciousness." But one of the main conclusions seems to be that decoding was still possible in a ton of areas (EVC, LOC, OPA, OFA, PPA, FFA), even for objectively invisible stimuli.

How does this finding match-up with the claims towards the end of the abstract? What are the implications for neural theories of consciousness, and how do these implications differ for obj-vis vs. obj-inv contrasts compared to subj-vis vs. subj-inv contrasts? The key question here is whether the neural results just change in magnitude or in kind when these different measurements of awareness are used.

2. Why does figure 3b not include obj-vis vs. obj-inv? A major focus of this paper is about how objective versus subjective measures of awareness lead to different neural results. Why only show subj-vis vs. subj-inv and subj-inv vs. obj-inv? Obj-vis vs. obj-inv seems just as interesting of a contrast for the purposes of this study.

3. On p. 8, first full paragraph, the authors state, "...participants did a simple one-back task on stimuli presented in a block design. This approach ... is often considered to isolate perceptual representations from later cognitive processes." I don't think this is the case. A one-back task requires subjects to hold each image in memory to see if it matches the next image. Holding things in memory seems like a "later cognitive process" and it's unclear how such a task isolates perceptual representations from memory processes.

4. Small but important typo on p. 5, second to last sentence in first paragraph of results: "(subj-vis: mean contrast 7.8%, SD 4.4; subj-vis: mean contrast 11.9%, SD 4.3)." I believe the first label should be "subj-inv".

Reviewer #3: This study is an important and timely attempt to compare neural activations evoked by stimuli that are rendered invisible, and that are either subjectively or objectively invisible. I absolutely love the research question and could not be more enthusiastic about it and about the importance of this study. I am no fMRI expert, but to me the analyses seemed compelling and comprehensive, and the introduction and discussion were well built, interesting, and provided a good description of the literature. The study also included a large sample size, which is highly important, especially for the small effects that are typical of the field. So my overall impression of this study is very positive. 

However, and unfortunately given my overall enthusiasm about this manuscript, I am concerned that the way objective and subjective invisibility were defined are not optimal, and increase the chances of finding differences between objective and subjective invisibility, as they lower the signal for the objective case and allow a stronger signal (or contamination by conscious perception) for the subjective case. Thus, the reported differences might not be an accurate estimation of the actual differences between objective and subjective invisibility, when operationalized correctly (see below). I would have loved to see a study where objective invisibility was determined per individual, and subjective invisibility would have indeed been defined as the lowest PAS level (which the authors claim is the case, but I disagree, as I explain below). I cannot understate how much I think such a study is important. In the absence of such a clean comparison, I would advise that the authors substantially tone down the claims about the differences between the two types of invisibility, acknowledge the limitations of their design, and instead emphasize the finding of unconscious processing even under the strict conditions of objective invisibility - a finding that is highly important (see point 3 below). To summarize, I find the study important and interesting, and would urge the authors to frame their results differently given the concerns I've raised. 

Major comments:

1. My first concern relates to the objective condition - typically, experimenters try to reach the sweet spot where the physical signal is as strong as possible, while objective performance is at chance. Although this was done here, to some extent, in the pilot experiment probing masking efficiency, it was not done well enough, I am afraid. The authors chose the maximal contrast of the mask based on the group performance in the pilot study Could it be that this manipulation is too strong, and substantially abolishes the signal evoked by the stimuli? A much preferable approach, which is often taken by researchers (e.g., Hesselmann et al., 2006, JoV) is to first calibrate the physical parameters (e.g., contrast) per subject, given the known differences between individuals, and find that sweet spot for each one. This is much more likely to allow higher-level processing also using the objective threshold. To take the current approach ad absurdum, one could also simply present the critical stimuli at 0% contrast, get objective chance performance and no signal whatsoever… This is of course not the case here (and indeed effects were found), but the authors should still explain why they chose not to calibrate per participant to allow for the strongest possible signal in the objective condition, and acknowledge that this could be the cause for the difference they find between the two conditions. 

2. The above issue is even more problematic when taking into account the definition of subjective visibility, which acts in the opposite direction; if the objective threshold is too strict, the subjective one seems too lenient. The authors write that subjects were instructed to report the stimuli as visible "even when they had only a vague idea of the stimulus category", and accordingly, claim that "subjective invisibility on our binary scale corresponded to the lowest visibility level of more fine-grained visibility scales, such as the four-point Perceptual Awareness Scale". I am not convinced that this is accurate, as the second level in PAS is "A feeling that something has been shown. Not characterised by any content, and this cannot be specified any further" (Ramsøy Overgaard, 2004). Thus, having a vague idea of the stimulus category is actually more visible than the second level, where one should have *no* access to any content about the stimulus - they only know something was there, but don't have any knowledge about what it was, not even a vague idea. This suggests that the "invisible" rating in this study actually reflects visibility levels 1 and 2, and not only 1. Unfortunately, this does pose a substantial problem, as the entire claim made here is based on the difference between subjective and objective threshold. But this comparison is now more problematic, as we are no longer comparing the lowest subjective invisibility with objective invisibility. I am not sure how this can be solved. To the very least, the authors should acknowledge this limitation and change the claims so to refer to the two lowest levels of subjective invisibility. The latter have been used in some studies to define invisibility (e.g., Melloni et al., 2011, J Neuroscience), but I wouldn't say it is the common practice in the field - typically, only the first level is used. Importantly, in many of these studies subjects are also at chance for the objective measures for visibility 1 trials - which is not the case here (and this might be explained by the fact that their "invisible" condition is not really invisible). Thus, I am not sure the critical test presented here - comparing the level of processing between objective and subjective invisibility - is a fair one. 

3. I agree with the authors that this work is probably the most robust demonstration of unconscious processing in fMRI to date. Given the concerns I've raised above, I would emphasize this point more, and deemphasize the differences between the objective and the subjective conditions (as again, I think the comparison here is biased towards finding differences). The unconscious findings by themselves are highly interesting and important, and also set the bar for future studies aiming to make similar claims. 

4. Was any correction for multiple comparisons across the paper? Such a correction is in order (e.g., FDR). The latter seemed to have been applied on the searchlight analysis, but there were many others conducted throughout the manuscript that should also be corrected for (for an explanation about the importance of correcting comparisons throughout the paper, see Benjamini et al., 2001, Behavioral Brain Research).

5. When talking about objective measures potentially underestimating unconscious processing, the authors should also mention as one of the reasons that above-chance performance in the objective task might be driven by unconscious effects.

6. The authors rightfully note that their sample size is larger than the typical fMRI studies in the field. But how was it determined? Was there any rationale behind it?

Minor comments:

1. I find it somewhat strange that when talking about "all major theories of consciousness", IIT is not mentioned. If the authors think it does not assume unconscious processing (which I don't believe is the case), they should change the text to "most major theories of consciousness". Otherwise, I think it should be cited there as well.

2. I apologize but I don't understand what is depicted in the line plots in figure 2. Are these simply lines connecting between the "peaks" of the bar plots? I don't really see why they are needed, as the readers can simply imagine them when looking at the bars (they are also somewhat misleading, as the variable is not continuous). I would remove them (or, if the authors really think they are necessary, add them to the bars plot. But I would advise to remove). If this is not what they depict, then please explain what is represented there.

---

## [Decision Letter · Decision Letter 2]

19 Apr 2021

Dear Dr Stein,

Thank you for submitting your revised Research Article entitled "Processing of subjectively and objectively invisible stimuli in human visual cortex" for publication in PLOS Biology. I have now obtained advice from the original reviewers and have discussed their comments with the Academic Editor. 

Based on the reviews, we will probably accept this manuscript for publication, provided you satisfactorily address the remaining points raised by reviewer 3. Please also make sure to address the data and other policy-related requests listed below my signature.

Editorially, we would like to make your title a bit more instructive, rather than descriptive, and suggest using: “The human visual system differentially represents subjectively and objectively invisible stimuli.” We would be happy to work with on a similar alternative.

We expect to receive your revised manuscript within two weeks. 

*Published Peer Review History*

*Early Version*

Sincerely,

Gabriel Gasque, Ph.D.,

Senior Editor,

ggasque@plos.org,

PLOS Biology

ETHICS STATEMENT:

-- Please indicate within your manuscript is your experiments were conducted according to the principles expressed in the Declaration of Helsinki or any other national or international ethical guideline.

DATA POLICY:

-- Thank you for uploading your quantitative data to OSF. I could not find files for Fig S1, even when the figure legend indicates the data should be there. Could you please either explain or upload those data as well? Please disregard this point if the data were already uploaded but I missed them.

DATA NOT SHOWN?

Reviewer remarks:

Reviewer #2: The authors have addressed all of my concerns (and in my opinion, the other reviewers' concerns as well). I am happy to support a positive decision on this version of the manuscript.

Reviewer #3: The revised manuscript addresses all my concerns, and I would be happy to see it published. Here are a few minor suggestions which might strengthen the manuscript, in my opinion, and make it more accurate:

1. The running title is "subjective vs. objective measures of awareness", which in my opinion underrepresents the scope of the paper. I would change to: "processing subjectively vs. objectively invisible stimuli".

2. The addition of a disclaimer about the lack of individual calibration is excellent, but I wonder if readers who are not experts in the field would understand the key message. I accordingly suggest adding an explanation at the end. After "although we cannot exclude the possibility that our group-based calibration approach did not result in optimal stimulus strength for every individual observer", I would add: "This might have reduced the chances for finding higher-level processing also for objective-invisibility", or "This might have inflated the difference between objective invisibility and subjective invisibility in our study".

3. The figures were of low quality; I imagine that this is just some technical limitation of the review process, but write this warning to make sure this would not be the case for the published paper.

---

## [Editor Report · Decision Letter 3]

20 Apr 2021

Dear Dr Stein,

On behalf of my colleagues and the Academic Editor, Rafael Malach, I am pleased to say that we can in principle offer to publish your Research Article "The human visual system differentially represents subjectively and objectively invisible stimuli" in PLOS Biology, provided you address any remaining formatting and reporting issues. These will be detailed in an email that will follow this letter and that you will usually receive within 2-3 business days, during which time no action is required from you. Please note that we will not be able to formally accept your manuscript and schedule it for publication until you have made the required changes.

***IMPORTANT: Please also make sure to add, within your manscript, an approval/ID number for your protocol approved by the University of Amsterdam ethics committee. (Please accept my apologies; I should have requested this information before).

PRESS

Thank you again for supporting Open Access publishing. We look forward to publishing your paper in PLOS Biology. 

Sincerely, 

Gabriel Gasque, Ph.D. 

Senior Editor 

PLOS Biology